# Complex interactions in the life cycle of a simple parasite shape the evolution of virulence

**Luís M. Silva**[1,2*], **Jacob C. Koella**[1]

**1** Institute of Biology, University of Neuchâtel, Neuchâtel, Switzerland, **2** Department of Zoology, University of British Columbia, Vancouver, Canada

\* luis.silva@ubc.ca

## Abstract

Evolutionary expectations about the virulence of parasites (i.e., the parasite-induced mortality rate of the host) often focus solely on the within-host transmission stage, overlooking the time spent between hosts and variations in transmission cycles. Moreover, the parasite growth rate within the host is closely linked to virulence. Here, we suggest that a simplified view of transmission and parasite evolution makes predicting how virulence will evolve difficult. We illustrate our ideas with a parasite with a simple life cycle, the microsporidian *Vavraia culicis*, which infects the mosquito *Anopheles gambiae*. We selected the parasite over six host generations for early or late host transmission, corresponding to shorter or longer time within the host. Selecting for late transmission increased their exploitation of the host, resulting in higher host mortality and a shorter life cycle with rapid infective spore production, comparatively to selection for early transmission. In response, hosts infected with late-selected spores shortened their life cycle and shifted to earlier reproduction. Using different host harm metrics, we demonstrate and discuss the pros and cons of using different measures of virulence. These and other findings emphasize the importance of considering the entire transmission cycle in studies of parasite evolution and raise concerns about how host density and social settings might influence virulence evolution.

## Author summary

Classical expectations on parasite evolution assume a trade-off between transmission rate and harm to the host (virulence), favoring parasites with intermediate virulence. However, recent studies challenge this idea, suggesting it is too simplistic. Here, by selecting the parasite *Vavraia culicis* for early or late transmission (or shorter and longer time within the host) in the host *Anopheles gambiae,* we demonstrate that understanding virulence evolution requires considering the entirety and the type of transmission cycle and not merely the within-host

**Data availability statement:** All relevant data are within the paper and its Supporting information files.

**Funding:** The study is supported by Schweizerischer Nationalfonds zur Förderung der Wissenschaftlichen Forschung (310030_192786 to LMS). The funders had no role in study design, data collection and analysis, decision to publish, or preparation of the manuscript.

**Competing interests:** The authors have declared that no competing interests exist.

stage. Moreover, we take a multi-fitness trait measure of virulence to fully understand the evolutionary outcome of the selection process. These findings highlight the importance of including all transmission stages in parasite evolution studies and demonstrate how certain social conditions might unintentionally select more virulent parasites.

## Introduction

Virulence is generally defined as the degree to which a parasite reduces the fitness of its host [1], commonly measured as the cost in host survival. It is the consequence of complex interactions between host and parasite strategies [2]. On one hand, the host can influence virulence through its ability to clear the parasite (its resistance) or to limit its effects and associated costs (its tolerance) [3,4]. On the other hand, the parasite's influence on virulence can be decomposed into processes that are linked to using its host's resources and growing (exploitation) or mechanisms that damage the host independently of the parasite's growth, such as toxins (per-parasite pathogenicity) [2,5,6].

Despite the complexity of this interplay [7], most ideas about the evolution of virulence are based on simple notions: that virulence is governed by the parasite and that its evolution is constrained by its trade-off with the rate of transmission. Thus, to maximize a parasite's total transmission (that is, the rate of transmission times the period of infection), evolutionary pressure should favor a higher rate of transmission until the advantage of faster transmission is balanced by a shorter infection period due to greater virulence. Therefore, evolution should lead to an intermediate level of virulence [8,9]. While this and variations of this idea have been extensively discussed and studied for the last 50 years [8–10], empirical studies have given mixed support [11,12]. This is partly due to an overly simplistic description of transmission, the key parasite fitness measure and a vital trait for infection development and spread. As a parasite fitness trait, transmission is rather complex, since it encompasses stages within and among hosts, each with their own set of factors that can interact and trade-off against each other [13]. Nevertheless, several of these factors have been studied, and indeed whether transmission is vertical or horizontal [14], continuous or not [15], or whether a parasite is directly transmitted, has environmental stages [13], or is vector-borne [16] strongly affects predictions about the evolution of virulence. However, a strongly overlooked factor is the timing of transmission, that is, the time it takes for the parasite to grow, develop, and potentially evolve within its host before being transmitted.

According to the trade-off model, earlier transmission should lead to higher virulence [17], though details of the timing of transmission and pathogenicity can lead to more complex predictions. One study that measured the effect of time to transmission was performed by Cooper and colleagues on nuclear polyhedrosis virus (NPV) infections in the moth *Lymantria dispar* [18]. They selected the virus for early or late transmission and tracked over generations the virulence and viral growth at two-time

points of the moth development. Selection for early transmission led to greater virulence but lower viral burden than selection for late transmission. While this result appears to agree with the virulence-transmission trade-off theory, it is difficult to interpret, for the experiment selected not only for early or late transmission but also inadvertently for rapid growth. This situation is made more complex by the fact that a longer time within the host before transmission might provide an opportunity for competing parasites to evolve [19,20]. Such competition is thought to be responsible for the evolution of higher virulence of the microsporidian *Glugoides intestinalis* when its host, *Daphnia magna*, is allowed to live longer [21]. Additionally, part of the parasite's impact may be due to the phenotypic responses of the host, which are governed by trade-offs between early and late reproduction.

Lastly, most studies that aspire to understand parasite evolution tend to focus on one transmission stage, either within or among hosts, and neglect the link between the two. This raises major concerns for epidemiology and the understanding of parasite and virulence evolution [13]. As expected from life history theory, a high fitness in one environment is likely to trade-off with success in other environments. This is particularly crucial for parasites, as their fitness is dependent on the success in the first host, survival in the environment they are transmitted to (e.g., water, soil, surfaces), and the ability to infect a new host. Hence, it is important to relate the diverse environments a parasite is exposed to fully understand its fitness and evolution. Focus on merely one of them might greatly bias our predictions regarding parasite evolution and which constraints between traits might exist. For instance, a parasite with high fitness within one host might not be as well equipped to survive the outside host environment, and vice-versa. Such trade-offs have been hinted at by some authors but in poorly controlled conditions.

Taking this into account, we first aimed to test whether selecting for the parasite's time to transmission affects its virulence. Next, in a common garden experiment, we examined if changes in virulence due to parasite evolution influenced infection dynamics and the response of non-evolved hosts. We then integrated our results into the modern virulence decomposition framework, which allowed us to separate the parasite's impact on the host into growth-dependent (exploitation) and growth-independent (per-parasite pathogenicity) costs. Finally, we connected these findings to a parallel study with the same parasite lines, which assessed differences in parasite endurance and infection ability after exposure to various temperatures and durations outside of the host.

For this, we used the mosquito *Anopheles gambiae* and its microsporidian parasite *Vavraia culicis*, which is a common generalist parasite of mosquitoes with low costs to the host [22–25] (Stock infection leads to a mean time to death of 18 days, compared to uninfected with 24 days) and whose life cycle allows us to easily control and manipulate both time-to and mode-of transmission. With this approach, we hoped to shed light on the role and importance of considering time to transmission, as well as within and among hosts' evolution in evolutionary and epidemiological studies. Results from this study are of universal interest for the understanding of parasite evolution and disease spread, but are also particularly relevant in vector-borne disease evolution, where most vector-transmitted parasites require long-lived hosts or vectors to complete their life cycle.

## Results

In this study, we selected the microsporidian parasite *V. culicis* for early or late transmission (and shorter or longer time within the host) in the host *Anopheles gambiae* for six generations (Fig 1a). Then, we assessed changes in parasite development within the host, as well as the host response to the parasite, when infected by the early or late-selected parasite, or upon infection with the reference stock parasite that did not undergo selection. Virulence of the infection by the different parasites was also measured and decomposed into exploitation (cost dependent of parasite growth) and per-parasite pathogenicity (cost independent of parasite growth). It is noteworthy to emphasize that virulence in this study is measured as a cost in host survival.

Concerning the parasite, we quantified its: i) cost in host survival (Fig 1b and 1c); ii) cost in host fecundity (Fig 1d); iii); costs in developmental traits (Fig 2); spore production rate and spore load dynamics (Fig 3); iv) compared virulence using

PLOS Pathogens

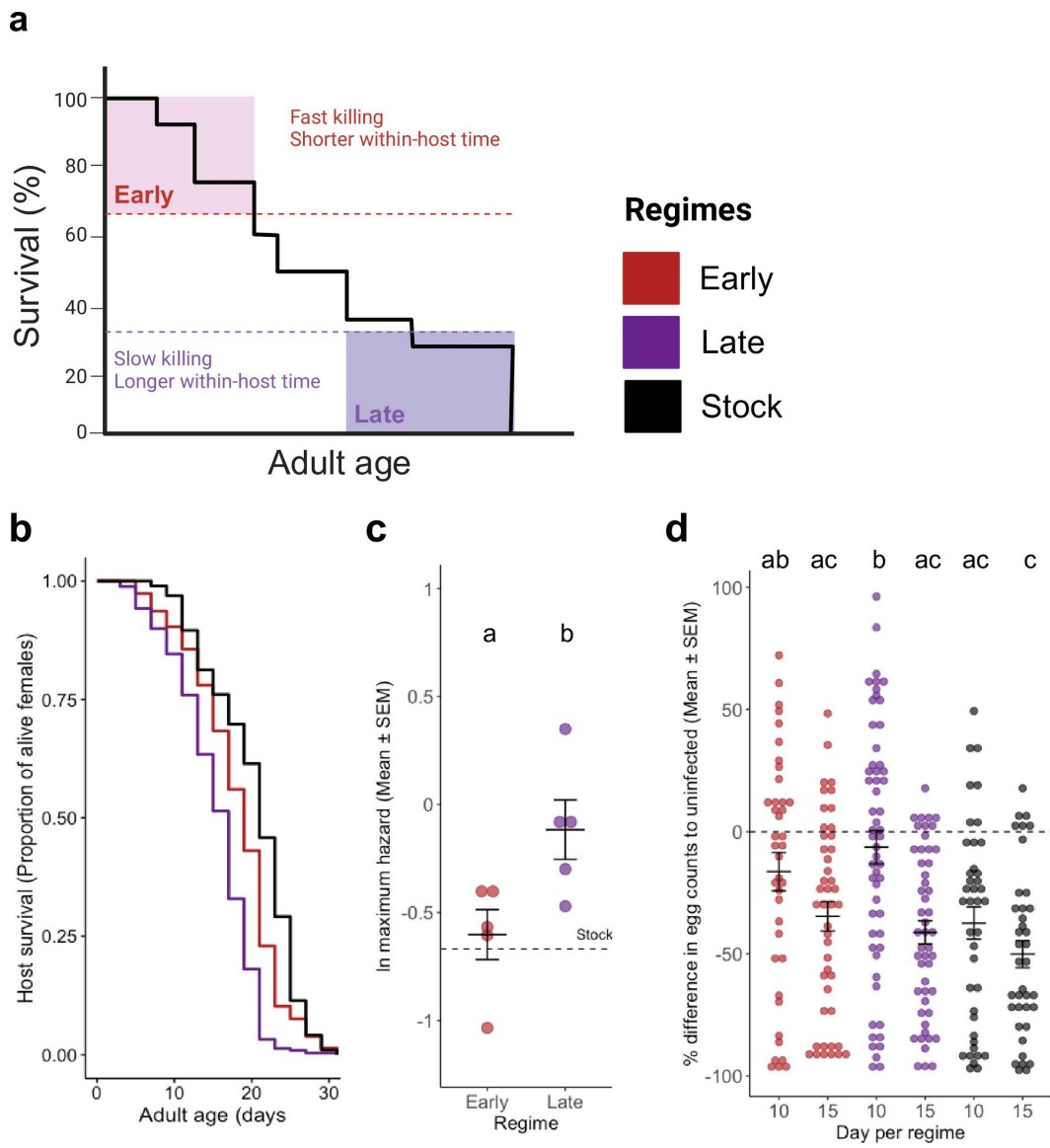

**Fig 1. Longevity, fecundity and virulence evolution.** (a) Experimental design, describing direct and indirect forms of parasite selection in this study. (b) Host survival after infection with evolved or reference parasites. Early and Late curves were generated with 487 and 519 females split throughout five replicates. The stock was composed of 96 females. (c) Virulence, as the natural log of the maximum hazard for each of the replicates of each selection regime ($n = 5$). The dashed line represents the virulence of the parasite stock. See Table A in S1 Text for a complete statistical analysis. (d) Cost of infection in fecundity, as the percentage difference in egg numbers of infected mosquitoes comparatively to the uninfected ones, for days 10 and 15 post adult emergence. The dashed line corresponds to the reference fecundity or uninfected individuals. The sample size was the following: Early with 39 and 45, Late with 56 and 50, Stock with 40 and 38 for days 10 and 15, respectively.

different host harm metrics (i.e., host larval, pupal and adult stage mortality (Fig 2), and fecundity (Fig 1d) to understand the infection fully (Fig 4); and v) using host mortality as the main proxy for virulence, we decomposed the parasite contribution for this virulence metric through the quantification of cost dependent and independent of growth in host survival - i.e., host exploitation and per-parasite pathogenicity, respectively (Fig 4). Regarding the host, we measured: i) its fecundity shifts in response to the different parasites (Fig 1d); and ii) developmental time and mortality during the larval and pupal

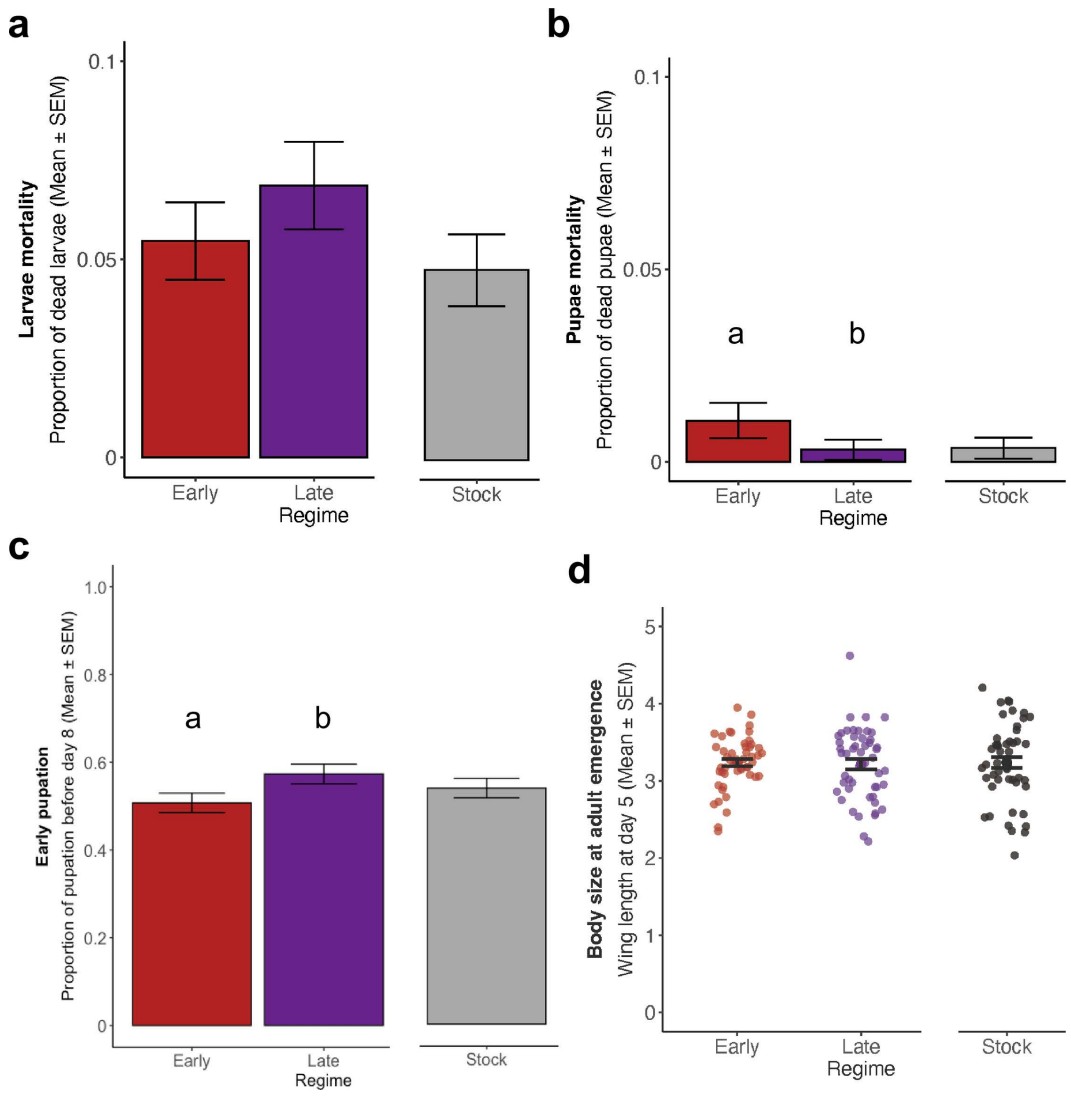

**Fig 2. Host developmental traits.** Selected regimes (i.e., early and late) were analyzed together with and separately from thetheirock. **(a)** Larvae mortality across regimes. The regimes were composed of 2068, 2011 and 2037 individuals for Early, Late and Stock. **(b)** Pupae mortality with each regime consisting of 1955, 1873 and 1936 in the same regime order. **(c)** Early pupation, as the proportion of individuals that pupated until day eight post eclosion. In the same regime order, the sample size was 1934, 1867 and 1929 individuals. **(d)** Body size at adult emergence using wing length of five-day-old females as a proxy. Each regime consisted of 50 individuals. Different letters denote means that they are significantly different from one another. See Table B in S1 Text for complete statistical analysis and Fig A in S1 Text for mean and standard error for each of the replicates.

stage upon infection (Fig 2). Altogether, with this set of results, we aimed to understand the drivers of parasite and virulence evolution within the host stage and how they contribute to host infectivity.

### Selection for late transmission increases virulence and shifts host fecundity

The selection regime had a cost in host longevity, with the individuals infected with spores selected for late transmission living less than individuals infected with early-selected or unselected spores ($\chi2 = 138.82$, $df = 2$, $p < 0.001$; Fig 1b). The comparison of the maximum hazard (a proxy for virulence) from the replicates of each regime gave a similar result, with

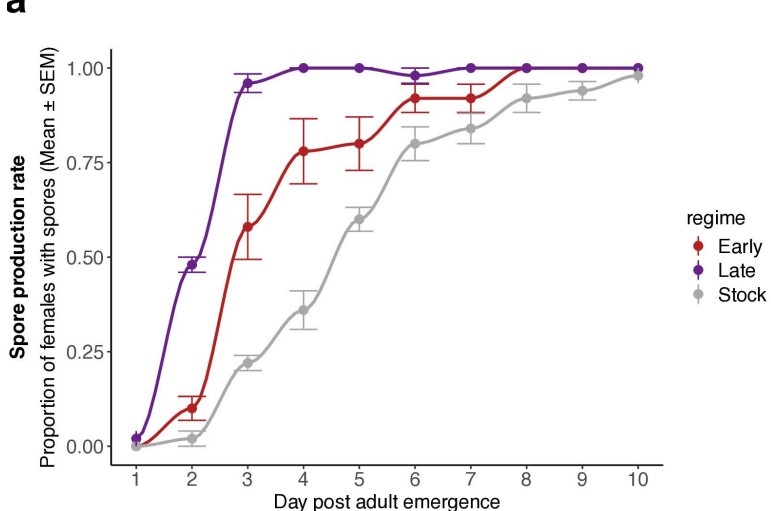

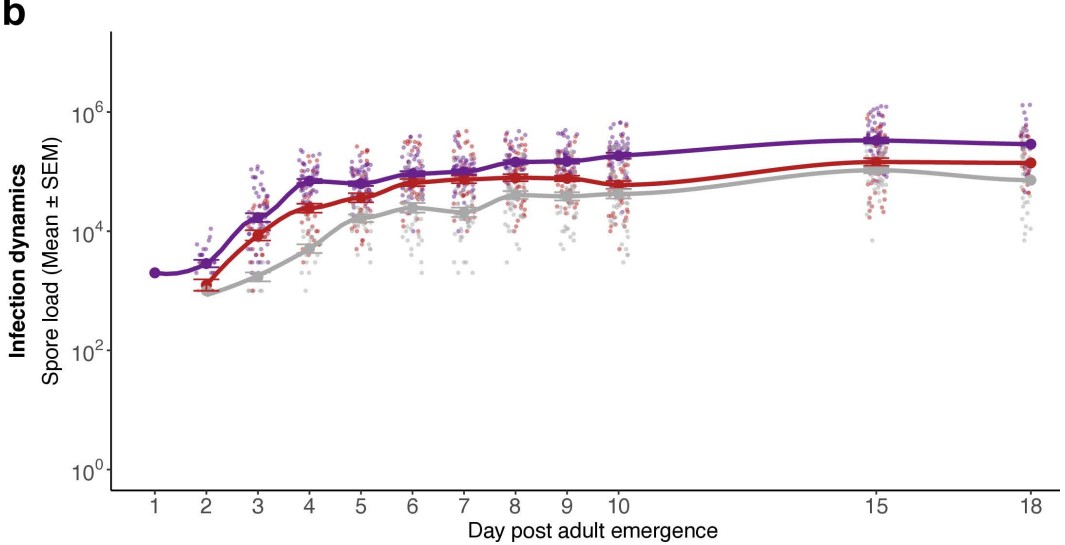

**Fig 3. Infection dynamics. (a)** Spore production rate, as the daily proportion of females with detectable spores for each treatment during the first ten days of adulthood. **(b)** Spore dynamics, as the mean number of spores per female for each of the first ten days, as well as days 15 and 18 (i.e., chronic load). For each regime and time-point, 50 females were assayed. See Table C in S1 Text for complete statistical analysis.

the infection from late-selected spores leading to an increase in maximum hazard and therefore virulence, using this fitness metric ($\chi2 = 13.239$, $df = 1$, $p < 0.001$; Fig 1c). Both these results suggest that selection for later transmission, and consequently longer time within the host, selects for more virulent parasites which greatly reduce the lifespan of the host, in comparison with infection by early-selected or unselected spores (Fig 1b and 1c). See Table A in S1 Text for further details on these analyses.

When accounting for the cost of infection in fecundity, we observed this was strongly affected by the selection regime as well ($df = 2$, F = 5.914, $p = 0.003$; Fig 1d). However, in this case, infection by both early- and late-selected parasites led to a lower cost for the host in comparison to infection by the unselected reference parasite. Nevertheless, the cost of infection in host fecundity was also affected by day ($df = 1$, F = 20.280, $p < 0.001$; Fig 1d), with the cost increasing from

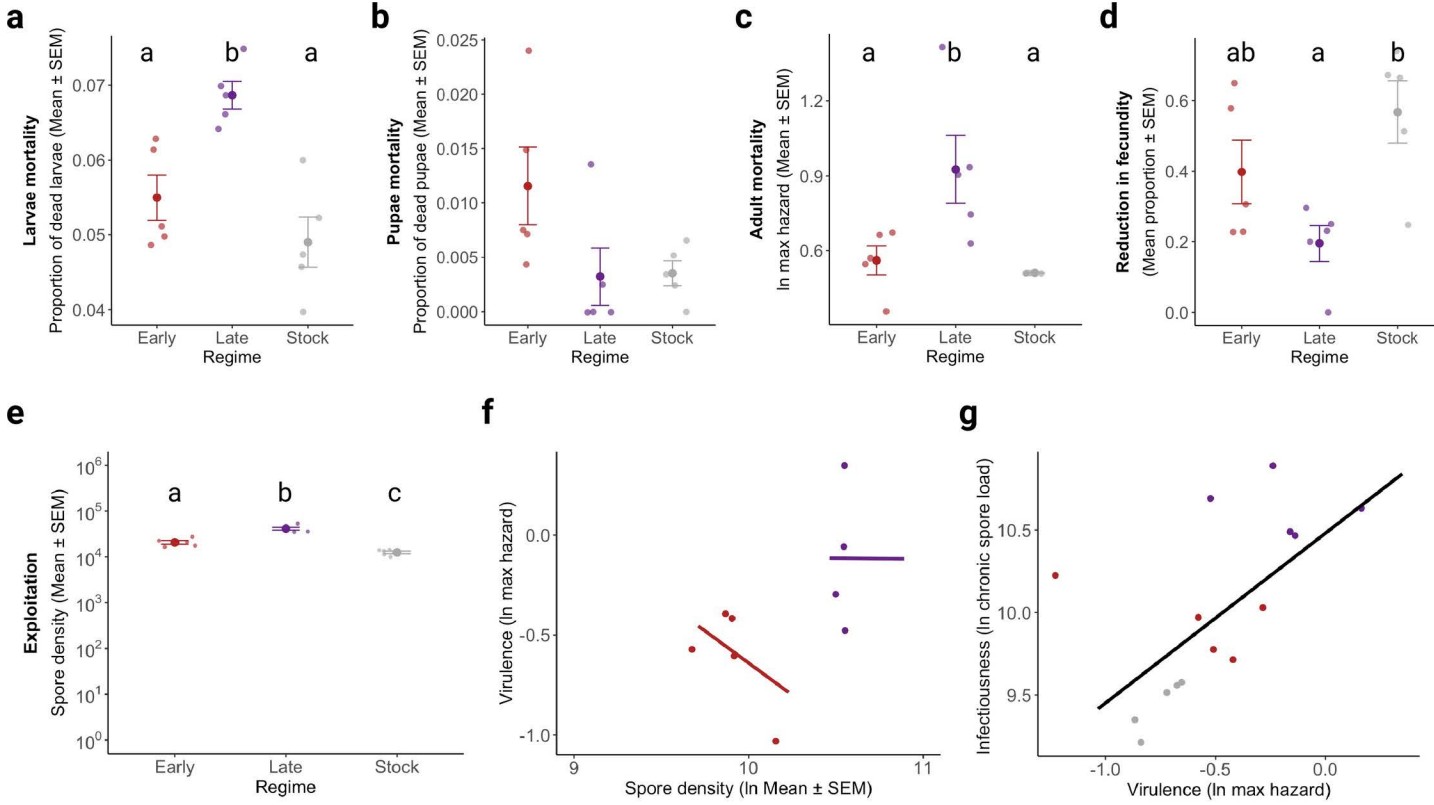

**Fig 4. Virulence metrics and decomposition.** Virulence was assessed using different metrics of host fitness/harm: **(a)** larval mortality, **(b)** pupae mortality, **(c)** adult mortality, and **(d)** reduction in fecundity (after conversion to a proportion, and the resulting value subtracted to 1). A linear model with regime as a fixed factor and replicate as a random factor was used to test for differences due to parasite treatment for each host fitness trait. **(e)** Exploitation was measured as the mean spore density at days 15 and 18 (i.e., spore load divided by wing size) for every replicate of each regime. A post hoc multiple comparison test was run afterward to determine pairwise differences between parasite regimes, illustrated by letters above the plot. If letters are missing, then no parasite regime differs from each other. **(f)** Per parasite pathogenicity, as the relationship between virulence and spore density. **(g)** Infectiousness-virulence relationship for this infection model. Given that in this infection model, the spore load is expected to decelerate in the measured chronic phase, the number of spores is not expected to increase and, therefore, provide a fair estimate of the total number of spores each of the regimes can transmit. There is a positive relationship between virulence and host infectiousness. See Table D in S1 Text for further details on the statistical analysis.

day 10–15 of host adulthood. Although there was no significant effect of an interaction between the selection regime and assay day, a multiple comparisons test was still performed to measure differences between each treatment for each day. Hence, we observed that the cost in fecundity in individuals infected with late-selected parasites on day 10 was less severe than either that of individuals with late parasites on day 15 of adulthood ($df = 258$, t-ratio = 4.164, $p < 0.001$) or unselected parasites on day 10 ($df = 258$, t-ratio = 3.533, $p < 0.001$) and 15 ($df = 258$, t-ratio = 4.916, $p = 0.006$). In sum, these results suggest infection by selected parasites had a lower cost in fecundity in comparison to infection by the unselected one. Also, the cost in fecundity increased with infection progression, in particular for individuals infected with late-selected parasites, where we observed an increase in fecundity for the first day, suggesting a shift in fecundity by the host. Further statistical details on Table A in S1 Text.

Considering virulence can be measured using different fitness traits, we tested for virulence across four different traits: stage-by-stage host mortality and fecundity (Fig 4). Interestingly, we found that although parasites selected for late-transmission increase adult- and larval-host mortality-inferred virulence (Fig 4a and 4c), they show different results

when considering fecundity for one egg clutch (Fig 4d). These findings demonstrate the selection protocol selected for: late-transmission parasites to invest in killing, while early-transmission ones to invest in host sterilizing, adding a new view on the evolution of infection.

**Selection shortened host and parasite life cycles**

Afterward, we focused on parasite-driven changes in host development (Fig 2). First, we compared larvae mortality between selected and unselected parasites, and then between early and late-selected parasites. We did not observe any statistical difference for either of the comparisons, showing parasite selection does not affect larvae mortality (Fig 2a). However, at the pupal stage, we observed that a host infected by an early-selected parasite was more likely to die in this stage, comparatively to hosts infected by late-selected parasites ($\chi 2 = 7.890$, $df = 1$, $p = 0.005$; Fig 2b). Nevertheless, selection for late transmission favored parasites that shortened the host time to pupation, when compared to early transmitted ones ($\chi 2 = 14.267$, $df = 1$, $p < 0.001$; Fig 2c). Although the last result provides evidence of a parasite-driven shortening of host developmental time, such outcome had no consequence in adult host body size ($\chi 2 = 0.054$, $df = 1$, $p = 0.816$; Fig 2d). We believe the shortening of the host cycle is considerably small and therefore, does not substantially affect the host growth. Further details can be found in the Table B in S1 Text, and replicate variation is presented in Fig A in S1 Text.

We then turned our focus to changes in infection parameters, such as spore production rate (i.e., the number of infected females with detectable spores, and our proxy for when the infective stages are produced) and their respective number produced per day (severity of infection and individual infectiousness). First, the spore rate increased with time ($\chi 2 = 114.375$, $df = 1$, $p < 0.001$) as it is expected. However, this increase was stronger for parasites selected for late transmission and weaker for unselected parasites (interaction between selection regime and day: $\chi 2 = 24.854$, $df = 2$, $p < 0.001$) (Fig 3a), meaning late-selected spores shortened their life cycle and started producing infective spores much sooner than the other treatments. The number of spores produced per day was also affected by the selection regime ($\chi 2 = 41.857$, $df = 2$, $p < 0.001$), day ($\chi 2 = 158.934$, $df = 1$, $p < 0.001$) and their interaction ($\chi 2 = 10.993$, $df = 2$, $p = 0.004$). Thus, the average number of spores produced increased with the progression of infection and did so more strongly if the parasites had been selected for late transmission (Fig 3b). Additional details are in Table C in S1 Text.

**An increase in host exploitation underpins virulence evolution**

We then aimed to relate the latter changes in infection dynamics to the parasite's respective virulence. For this we decomposed virulence into two components: a) exploitation, i.e., cost in host survival due to parasite growth; b) per parasite pathogenicity, i.e., cost in host survival independent of parasite growth. This decomposition allows us to understand if parasite evolution is due to changes in the parasite growth rate, or pathogenicity, such as the acquisition of virulence factors or toxin production.

In our case, exploitation (i.e., cost due to parasite growth) was affected by the selection regime ($df = 2$, F = 55.466, $p < 0.001$), but per parasite pathogenicity was not ($df = 1$, F = 0.302, $p = 0.602$) (Fig 4a and 4b; See Table D in S1 Text for statistical comparisons). These findings suggest that, although we did not detect a significant effect of selection for early transmission on virulence, it did increase the parasite's ability to exploit the host, though to a lesser degree than selected for late transmission. Considering *V. culicis* growth slows down around days 15 and 18 of adulthood, almost resembling a plateau, we used spore density as a proxy of individual infectiousness and tested for a relationship between infectiousness and virulence in this model system. It is important to mention that this sort of rationale is valid for the parasite model in question, as our measure of parasite growth is the number of infective cells that will be transmitted to the environment and consequently infect future hosts. Therefore, for this model, spore growth is equivalent to host infectiousness, one of the transmission stages discussed in Silva *et al.* (2025) [13]. Using a linear model, we detected an effect of virulence (i.e., natural log of the maximum hazard) on host infectivity (i.e., natural log of mean spore density) ($df = 1$, F = 31.534, $p < 0.001$,

Fig 4c, statistical details on Table D in S1 Text), providing evidence for a positive correlation between host infectivity, or spore growth, and virulence in this system.

## Discussion

We demonstrated that time to transmission affects the evolution of virulence. In particular, we showed that contrary to the predictions of a theoretical model [17], a longer time within the host (i.e., late transmission) was selected for greater virulence than a shorter time (i.e., early transmission).

Furthermore, we provided contrasting evidence to a study on nuclear polyhedrosis virus (NPV) infections in the moth *Lymantria dispar* [18], in which selection of the virus for late transmission led to lower virulence (and higher viral burden) than selection for early transmission. Below, we argue and explain why these two contrasting findings might reflect current limitations in the study of parasite evolution. We also demonstrate that our study serves as a suitable example to reflect on classical theories of parasite evolution and how the integration of parasite life history is vital for the understanding of its evolution and, consequently, virulence itself.

The different outcomes observed in these two studies emphasize the complexity of the evolution of virulence [7,12,13]. In particular, it may result from the different levels at which selection operates on parasites: i) competition between parasites within one host; ii) the ability to survive an environment outside the host; iii) or transmission among hosts [13]. The interlinking of these, and dependent traits, can either limit or facilitate their evolvability, which in turn has major consequences for the understanding of virulence evolution. Indeed, if we only consider selection at the level of transmission among hosts, as assumed by much of the theory of the evolution of virulence [8,9] and by Osnas and Dobson (2010), selection for early transmission would lead to a higher killing rate than selection for late transmission.

However, if we instead consider sources of selection within the host, we can expect a different evolutionary outcome, which in turn aligns with our results.

Since we selected for early and late transmission by letting the parasite be transmitted from mosquitoes that died early or that lived a long time, we indirectly selected for shorter and longer periods of infection. A longer period of infection, in turn, increases the chance of mutation (an unlikely explanation for our results) or the strength of density-dependence as the parasite load increases, and thus increases the possibility of competition among parasite genotypes [26,27], in a phenomenon described as clonal interference [19]. This selects for more transmissible or more rapidly replicating parasites in our study (Figs 3 and 4) and others [19,26]. This result is experimentally supported by a recent study from Sheen and colleagues (2024), which suggested that in the absence of turnover rates in poultry markets, and longer host lifespan, virulence is no longer expected to be maintained at intermediate levels but instead continuously increases as observed with circulating parasites such as H5N1 [28]. We expect this phenomenon to happen not only in poultry markets with lower turnover but also in society and the wild, and literature has reflected this trend [27]. Longer infection gives the immune response more time to act, giving greater selective pressure for the parasite to evade immunity and, consequently, replicate more rapidly and be more virulent [29], thus hindering the fixation of genotypes with low virulence [30]. In addition, the host's immune response should increase at higher parasite burdens [6,31]. Therefore, early-transmitted parasites, which have had little time to replicate, are subjected to less selective pressure from the host's immune response. Late-transmitted parasites, however, grow to a higher density and are subjected to a strong immune response, which then fades with the mosquito's age [32,33], letting the selected genotypes replicate rapidly. Overall, thus, in contrast to selection due to transmission among hosts, within-host processes are likely to select for high virulence of parasites that can replicate for a long time and that transmit late.

A good example of this rationale is provided by the evolutionary models of obligate killers [12,34–36], which assume that the host is a limited resource, making the parasite's growth density-dependent and decoupled from its transmission. This makes it possible for "selfish" genotypes to grow without counter-selection [34,37]. In our study, although *V. culicis* is not an obligate killer (see Materials and Methods), the experimental design made it behave like one by restricting

its transmission route, and therefore, both assumptions are likely to hold in our experiment. In particular, our protocol eliminated many costs of transmission. Thus, for example, since the spores (infective stage of our parasite) were sampled shortly after their production, transmission was independent of their long-term quality or infectivity, so any trade-off between the replication rate and quality of spores played no role in transmission [12,38,39]. Indeed, the positive correlation between chronic spore load (i.e., a proxy for infectiousness) and virulence (when results of all selection regimes and replicates are pooled) (Fig 4c) corroborates this uncontrollable growth hypothesis. Several studies and models have supported this hypothesis [5,40,41].

This hypothesis introduces us to the third level at which selection can act on: the ability to endure the environment among hosts, a transmission stage common to most parasites [13,40,42]. This resting stage is crucial for *V. culicis* because its transmission cycle encompasses the death of an individual on water and the release of infective spores that will be ingested by young larvae. These spores can sit and wait for a few hours to several months [43], making their survival in the external environment essential for a successful transmission. In respect to this, a follow-up study of ours with these same evolved lines quantified the parasites' ability to endure external conditions and their resilience to different temperatures over time [44]. Consistent with life history theory, we found that spore growth within the host was negatively correlated with survival in the external environment. This meant that spores selected for late transmission, which had the highest growth rate within a host, were the most susceptible to dying in the external environment. This effect was more pronounced with longer exposure to the outside environment and at higher temperatures. In contrast, early-selected and unselected reference spores remained highly viable for the whole duration of the experiment, i.e., 90 days. A reduction in parasite viability in late-transmitted spores led to lower infectivity in new hosts and less severe infections with lower parasite burdens compared to their counterparts. Hence, we demonstrated that selection within the host can inadvertently impact parasite fitness outside of the host. This is of particular significance from a social perspective, as increased host density due to scenarios with social crowding or aggregation behaviors might shorten the time parasites spend in the external environment. This favors parasites that prioritize evolution within the host stage and which invest in host exploitation and within-host growth.

Several other results in this study are noteworthy. First, adaptation to either early or late transmission increased the parasite's prevalence of spores (Fig 3a), growth (Fig 3b) and exploitation of the host (Fig 4a). This is most likely because the parasite switched from being maintained on two mosquito species as hosts to a parasite infecting only on *An. gambiae*. The parasite could thus specialize on a single host without being impeded by the cost of having to also perform on a second one [25,45,46]. However, selection did not affect per-parasite pathogenicity for any of the regimes (i.e., damage independent of parasite growth such as production of toxins, Fig 4b). This is likely due to the limited microsporidian genome [47,48] and, to our knowledge, lack of virulence factors on which selection can act.

Second, our selection protocol had an impact on some aspects of the mosquitoes' development. For instance, infection by late-selected parasites led to a shortening of the host life cycle through early pupation (Fig 2c). This result is likely due to the properties of this parasite treatment. Indeed it has been hypothesized infection by more virulent parasites induces strong stress responses that activate higher levels of immunity and shorten the host developmental time [6,49]. This hypothesis is supported by our results, as the most virulent parasite (late-selected) shortened the host developmental time (Fig 2c), while the less virulent parasite (early-selected) did not, and instead carried a cost in pupae mortality (Fig 2b). Nevertheless, neither of these changes was big enough to impact body size at adult emergence (Fig 4d).

Third, infection decreased fecundity, in particular later in life. This impact does not appear to be linked to the density of spores, for the average number of spores did not differ between the time points when fecundity was measured and infection by the unselected parasites gave the lowest fecundity despite having low spore loads (Fig 3). Rather the pattern may reflect the host's response to infection. Indeed, as suggested by the life-history theory [50], the shorter the lifespan of infected mosquitoes, the greater the early investment in fecundity, while late fecundity was low and similar in each selection regime.

Another important consideration of this study is the choice of the right metric of virulence. Depending on the infection setting, ecology, or host-parasite evolutionary history, it is often hard to decide which fitness or health trait to measure virulence and the harm caused by the infection to the host. A few authors have tackled this by measuring virulence across different fitness traits and calculating a somewhat cumulative virulence measure [51–53]. Here, using our host fitness measures, we did a similar comparison (Fig 4a–4d), which clearly showed that the two selection regimes selected for higher virulence in two different host metrics: while parasites selected for late-transmission invested in host mortality during the larval stage (Fig 4a) and predominantly adult stage (Fig 4c), early-selected parasites invested in sterilizing the host to their benefit, by reducing the number of offspring produced in a single clutch (Fig 4d). These findings fit nicely with the discussion points we just argued and illustrate the complexity of infection, and how focusing on the traditional metric does not always allow for a better understanding of the evolution of infection. Nevertheless, although we found differences in virulence depending on the fitness metric used, we do not believe it is appropriate to calculate a cumulative virulence measure, as we believe in a hierarchy of fitness and health traits, which are interlinked. Moreover, in addition to being interlinked, fitness traits often have case-specific impacts that depend on the environment and respective ecological constraints. This makes a cumulative measure challenging to calculate without such parameters. Nevertheless, it is evident to us that at least two host fitness traits are universally of the utmost importance for virulence evolution. Those are host survival and host reproduction, as although they are influenced by a variety of factors, they reflect the contribution to the next generation - which, in the end, is the most rigorous measure of fitness.

Altogether, our study highlights the importance of considering the whole parasite life cycle to understand its evolution. The findings and arguments presented here are intended to prompt the reader to reflect on classical theories and their applicability to modern society and disease spread strategies. We validate many of the concerns previously discussed in the literature concerning the current theory of virulence evolution and how they link to the oversimplification of parasite transmission, the main parasite fitness measure. Although usefully simplistic, the current theory of virulence disregards a considerable portion of parasite strategies, which encompass different transmission stages that evidently cannot be neglected, even when solely studying within-host aspects of infection. Importantly, we believe this study has major consequences for epidemiology, vector control, and disease spread strategies, regardless of the host-parasite model system in question. The latter is due to the nature of parasites' transmission cycles, which, as discussed in detail by Silva *et al.* (2025) [13], present the same transmission stages regardless of the taxa, and therefore, evolutionary outcomes and interactions discussed here can be adapted and further explored in most parasite taxa. Moreover, one of the key take-home messages is the importance of the balance between within-host and among-host evolution. This has obvious consequences for disease evolution and spread, as it has become increasingly obvious and alarming in recent pandemics, such as the SARS-CoV2, with the rise of disproportionately high transmission associated with higher virulence in certain social settings [54–56]. Only through the understanding of this balance can we predict and design better disease control strategies. Undeniably, a continued short-sighted view of parasite evolution with within-host evolution as the sole focus of the field will hinder our understanding of parasite evolution and promulgate the controversy between virulence and transmission.

Lastly, we believe is imperative to make a small nod to the importance of these findings to the ecology of this parasite. Microsporidia, such as the one in this study, have been proposed as biological agents for malaria control [22,57] due to their ability to suppress *Plasmodium* infections in wild- and lab-grown mosquitoes [58,59]. In line with this, it is possible that the geographical distribution of mosquitoes (and microsporidia they carry) might influence microsporidia evolution and therefore their control potential. Surely, infection by our microsporidia is unlikely to affect mosquito population numbers, as our late-selected treatment with shorter lifespan has great earlier fecundity (Fig 1d), but the same cannot be expected regarding wild mosquito density. Although the mechanisms underpinning *Plasmodium* suppression are still unknown, it is fair to expect microsporidia evolution to trade off with its ability to interfere with *Plasmodium* development to a certain extent. Hence, this system seems a unique model to test microbe-driven vector-control mechanisms, while allowing to study of fundamental evolutionary questions regarding infection and microbe interactions.

## Materials and methods

### Experimental model

We used the microsporidian parasite *Vavraia culicis floridensis*, provided by James Becnel (USDA, Gainesville, FL) and the Kisumu strain of the mosquito host *Anopheles gambiae (s.s)*. Mosquitoes were reared under standard laboratory conditions (26 ± 1ºC, 70 ± 5% relative humidity, and 12 h light/dark). *V. culicis* is a generalist, obligatory, intracellular parasite that infects mosquito epithelial gut cells. Mosquitoes are exposed to the microsporidian during the larval aquatic stage, where they ingest spores in the water and food. After replication within the larvae, they start producing spores from within the adult gut cells, re-infecting new cells and propagating the cycle until they kill the host. Spores are transmitted when infected individuals die in water or from mother to offspring by adhering to the surface of the eggs. Before starting the experiment, *V. culicis* was kept in large numbers by alternatingly infecting *Aedes aegypti* and *An. gambiae* populations to ensure it remains a generalist parasite. The infection protocol was performed as the experiment described below, with two-day-old larvae being exposed to 10,000 spores/ larvae. Dead infected mosquitoes were collected shortly after death and kept in the fridge at 4°C for later infections.

### Experimental evolution and rearing

A starting population of 300 freshly hatched *An. gambiae* larvae were grown in groups of 50 per petri dish (120 mm diameter x 17 mm) containing 80 mL of distilled water. Larvae were fed with daily Tetramin Baby fish food according to their age: 0.04, 0.06, 0.08, 0.16, 0.32 and 0.6 mg/larva for 0, 1, 2, 3, 4 and 5 days or older, respectively. On the second day of development, individuals were exposed to 10,000 spores of *V. culicis* per larva. Upon pupation, individuals were transferred to cages (21 x 21 x 21 cm) and left to emerge into adults. Freshly prepared cotton pads with 6% sucrose solution were provided throughout their adult life. At adulthood, two selection regimes were generated (Fig 1a): i) Early transmission, where the spores from the first 50 females to die were retrieved, mixed and used to infect the following generation of the same regime; ii) and late transmission, where the spores from the last 50 females to die were used. It is noteworthy to mention that we are aware that this selection protocol could favor mosquitoes with low fitness or especially susceptible to infection and, therefore, bias our selection for early transmission. However, based on previously published data on this host-parasite model, we have concluded this would only happen to a smaller extent, not affecting the overall evolution of the selection regime. The latter is because fragile mosquitoes with low fitness tend to die up to 48 hours of adulthood, regardless of infection. Infected individuals only start dying from day seven of adulthood. Considering this, we believe such constraint might not affect our results in any way, as: 1) we found one to five mosquitoes per replicate (out of 50) per generation to die within this time frame, barely contributing to the next generation of the parasite; and 2) as seen from Fig 3, mosquitoes that do survive up to day two of adulthood do not have any spores being released, and therefore they are evidently not contributing to the next generation. The same rationale can be employed for the late-transmission regime, where hosts that might be uninfected will also not contribute to the next generation of parasites. However, to our knowledge, in this experimental setup, all individuals are usually infected, and no uninfected individual has been found before the start of these selection lines.

   Each regime consisted of five replicate populations and was selected for six generations prior to assessing differences due to selection at F7. To ensure transmission only happened through parasite killing, dead individuals were removed shortly after death, and female mosquitoes were never blood-fed and therefore unable to reproduce. The retrieved dead females were kept in 2 mL microcentrifuge tubes with 1 mL of distilled water at 4°C for no longer than 40 days, to maintain the viability of the spores. At the end of each generation, we pooled all selected individuals, added a stainless-steel bead (Ø 5 mm) to each of the tubes and homogenized the samples using a Qiagen TissueLyser LT at a frequency of 30 Hz for one minute. The concentration of the resulting spore solution was then estimated by counting spores in a haemocytometer under a phase-contrast microscope (400x magnification) and the dose for the next generation was adjusted to 10,000

spores/larva. We only selected the parasite, and therefore, we used mosquitoes from the stock population to start every generation.

### Response to selection

After six generations of selection, we measured the effect of parasite selection on a set of host traits: larvae and pupae mortality, time to pupation, body size, fecundity, longevity, and infection dynamics for each of the *V. culicis* selection regimes and replicates. We included the "stock" *V. culicis* population as a regime as our baseline reference. All the common garden experiments were performed in a similar design to the selection protocol unless otherwise stated. Three experiments were run in parallel to measure these traits, and these are described below.

**Longevity and virulence.** Two-day-old larvae were infected with the standard dose of 10,000 spores/larva, as described above. The individuals were reared in groups of 50 larvae per petri dish per combinatorial treatment (selection replicate x parasite treatment). Upon pupation, pupae were transferred to cages and left to emerge into adults. Once they emerged, female adults were isolated in individual cups with a humid Whatman filter paper for humidity) and a 6% sucrose solution (for feeding and routinely changed to avoid contamination). Around 100 females were considered the sample size for each combinatorial treatment. Then, survival was tracked daily up to the death of the last individual.

**Fecundity.** This experiment was performed in a similar manner to the previous one, except that in this one, the adult females were allowed to mate. This experiment was split into two sub-experiments. One subgroup was allowed to mate immediately and reproduce in the first window (10 days of adulthood), while the other was allowed to mate five days after emergence to reproduce only in the second window (15 days) for the first time. This setup allowed us to control reproductive and carrying costs from one egg-laying to another. Females were allowed to mate in groups of 50 females and 50 males per cage (21 x 21 x 21 cm). While females were infected by a given parasite treatment, all males were of the same age as the females but uninfected and were grown in the parallel with the experiment to remove any paternal epigenetic effects. After seven days of mating, all the males were removed from the cages, and the females were allowed to blood-feed on Tiago G. Zeferino's arm for five minutes. Two days later, 100 females from each treatment and replicate were individualized in cups (5 x 5 x 10 cm) with Whatmann filter paper (110mm) and water and allowed to lay eggs for 48 hours. Afterwards, all females were killed, and the number of eggs counted.

**Spore prevalence, dynamics, and developmental traits.** This experiment started with approximately 500 mosquito larvae per combinatorial treatment, resulting in 2000 larvae per parasite treatment (i.e., early-transmission, late-transmission or stock). Larvae were infected, as described above. Both larvae and pupae mortality, as well as time to pupation, were assessed every 24 ± 1 h. Body size at adult emergence was measured using wing length at day 5 as a proxy [14]. Females were not allowed to mate and, therefore, were directly transferred after adult emergence to individual cups (5 x 5 x 10 cm) with a humid Whatmann filter paper (110mm) for humidity and a wet cotton with 6% sucrose solution for feeding. Infection dynamics, as in spore prevalence and load, was followed by randomly sacrificing 10 alive females per combinatorial treatment on days 1, 2, 3, 4, 5, 6, 7, 8, 9, 10, 15, and 18 of adulthood and then assessing if they had any spores (spore prevalence) and if so, how many (spore load). Both spore prevalence and load were quantified by counting spores in a hemocytometer under a phase-contrast microscope (400x magnification).

### Statistical analyses

All the analyses were performed with R version 4.3.1 in RStudio version 3034.06.2 + 561. The following packages were used for visualizing and plotting the data: "ggplot2" [60], "purrr" [61], "scales" [62], "dplyr" [63] and "tidyr" [64], as well as created with BioRender.com and edited in Microsoft PowerPoint 16.16.27. Statistical analysis and their diagnosis were conducted using: "DHARMa" [65], "car" [66], "survival" [67], "lme4" [68], "muhaz" [69], "emmeans" [70] and "multcomp" [71].

Longevity was compared across regimes using a cox proportional hazard with the regime as the explanatory variable and the replicate as a random factor. Then, we generated a survival curve for each replicate of each regime and extracted its maximum hazard, as in [6], our proxy for virulence. We used maximum hazard values as our proxy for virulence as this value provides the highest value instantaneous failure rate for a given survival curve and, therefore, represents an adequate measure of the virulence of a given infection. To extract maximum hazard values, we used a function that used the "muhaz" package [69] to generate a smooth hazard function and then output the maximum hazard, as well as the time at which this maximum is reached. Differences in virulence were assessed through a linear model with the natural log of the maximum hazard as a dependent variable, the regime as the explanatory variable, and the replicate as a random factor.

Concerning the cost of infection in fecundity, we first tested if the proportion of non-egglaying females was the same across all regimes and days for each replicate. For this, we ran a generalized linear model with a binomial structure and both regime and day post-adult emergence as fixed factors, as well as their interaction, and replicate as a random factor. There was no effect of any of the treatments and therefore, only considering females who had laid eggs, we calculated the percentual cost of infection by subtracting the mean uninfected fecundity from each infected individual fecundity, for day 10 and 15 of adulthood, and dividing the resulting value by the infected individual fecundity and multiplying it by 100. Then, we assessed differences due to regime and/or days post-adult emergence using a linear model with replicate as a random factor, and day of adulthood and regime as explanatory variables (as well as their interaction).

Larvae and pupae mortality were measured as the proportion of dead individuals, for each of the stages. A generalized linear model with a binomial error structure, and the replicate as a random factor, was used to determine the effect of regime on each of the developmental stage mortality. Individuals pupated from day 6–11 post eclosion, but approximately 77%, 79% and 82% of them pupated on day eight or nine, for Early, Late and Stock, respectively. Therefore, we split individuals into two categories, the ones that pupated until day eight and the ones that pupated from day nine onwards and used a generalized linear model with binomial distribution to measure differences in early pupation. As before, the regime was considered an explanatory variable and the replicate as a random factor. Differences in body size were quantified using two linear models: i) the first model tested for differences between infection by selected parasites (i.e., Early and Late) or unselected (i.e., Stock) using selection/no-selection as the explanatory variable and replicate as a random factor; ii) then, we tested for differences due to infection between selected parasites, meaning Early and Late regimes, using a similar model, with the parasite treatment as an explanatory factor and replicate as a random factor.

We defined spore rate as the proportion of females with detectable spores per regime for each of the first ten days of adulthood. Our detection threshold is estimated to be 100 spores. Mosquitoes with detectable spores were then used to estimate spore dynamics from day one to 18 of adulthood. The spore rate was analyzed using a generalized linear model with a binomial distribution, while for spore dynamics, we used a linear model to assess differences in the log-transformed number of spores. In both models, we had regime and day, and their interaction, as explanatory factors and the replicate as a random factor.

Since we measured several fitness traits for this infection system (i.e., stage-by-stage host mortality and fecundity), we assessed virulence using different fitness traits as proxies (Fig 4). For this, we considered the following traits: i) larval host mortality, as the proportion of dead larvae; ii) pupae host mortality, as the proportion of dead pupae; adult host mortality, as the maximum hazard of the adult survival curves generated in Fig 1; and iv) Impact in clutch fecundity, as the mean fecundity between the two time-points (10 and 15 days of adulthood) by subtracting the transformed fecundity values into proportions and subtracting them to one. We used a linear model to test the effect of regime (fixed factor) and replicate (random factor) on each virulence proxy. In the presence of a significant main effect, we ran a posthoc test to perform pairwise comparisons between our parasite regimes (Fig 4 and Table D in S1 Text).

To decompose virulence, we used adult host mortality as our chosen measure of virulence. The rationale for this decision is that the fitness traits measured seem to be the most impactful out of the different survival measures. Although we measured clutch fecundity, we cannot say for sure there is an impact on the lifetime reproductive output in this system, as

it is beyond the scope of the project. We then resumed virulence decomposition by using the maximum hazard from the longevity curves and (since the average maximum hazard day was 20) the spore density on days 15 and 18. Spore density was calculated as the individual spore load at days 15 and 18 adjusted by individual wing length. Virulence decomposition analysis was done as in [6]. In summary, exploitation was measured as the natural log-transformed mean spore density from days 15 and 18 for a given regime and replicate, while per parasite pathogenicity was given as the slope of the relationship between natural log-transformed maximum hazard and (i.e., virulence) and exploitation. Lastly, post-hoc multiple comparisons were performed with "emmeans", using the default Tukey adjustment, across the different models. Infectivity has been previously defined as the "number of parasites released from the first host to the next stage, which can be the outside environment or a new host", according to Silva *et al.* (2025) [13]. Hence, the chronic parasite load of each individual was considered a proxy of infectiousness as it reflects the amount of parasite that will be transferred to the environment in case of host death (particularly in water). The relationship between infectiousness and virulence was then analyzed using a linear model with infectiousness (or mean chronic load at days 15 and 18 of adulthood) as the dependent variable and the log-transformed maximum hazard of every parasite line as the independent variable.

## Supporting information

**S1 Text.** Table A. Longevity, fecundity, and virulence evolution. Longevity was analyzed using a cox proportional hazard with the different regimes as explanatory variables and the replicate as a random factor. Once the survival curves were generated, the maximum hazards for each of the survival curves were extracted using the "muhaz" package [69]. Maximum hazard served as our proxy for virulence, which was then compared across treatments using a linear model with the natural log of the maximum hazard as a dependent variable, while the regime was considered an explanatory factor and the replicate as a random factor. The infection cost in fecundity was calculated only for females who laid eggs (and therefore mated and had access to blood). We calculated the percentual infection cost by subtracting the mean uninfected fecundity from each infected individual fecundity for day 10 and day 15 of adulthood and then dividing the resulting value by the infected individual fecundity and multiplying it by 100. Then, differences due to regime and days post-adult emergence were assessed using a linear model with regime and day post-adult emergence as explanatory variables, as well as their interaction. Moreover, we also included the replicate as a random factor. Further details can be found in the "Statistical analyses" section of the Materials and Methods. Table B. Host developmental traits. Larvae and pupae mortality were assessed as the proportion of dead individuals for each stage. A generalized linear model with a binomial error structure and the replicate as a random factor was used to determine the effect of the regime on each of the developmental stage mortality. Individuals pupated from day 6–11 post eclosion, but approximately 77%, 79%, and 82% of them pupated on day eight or nine, for Early, Late, and Stock, respectively. Therefore, we split individuals into two categories, the ones that pupated until day eight and the ones that pupated from day nine onwards, and used a generalized linear model with binomial distribution to measure differences in early pupation. As before, the regime was considered an explanatory variable and the replicate as a random factor. Differences in body size (using wing length as a proxy) were quantified using two linear models: i) the first model tested for differences between infection by selected parasites (i.e., Early and Late) or unselected (i.e., Stock) using selection/no-selection as the explanatory variable and replicate as a random factor; ii) then, we tested for differences due to infection between selected parasites, meaning Early and Late regimes, using a similar model, with the parasite treatment as an explanatory factor and replicate as a random factor. Further details can be found in the "statistical analyses" section of the materials and methods. Table C. Infection dynamics. The spore production rate was defined as the proportion of females with detectable spores per regime for the first ten days of adulthood. Mosquitoes with detectable spores were then used to estimate spore dynamics from day one to 18 of adulthood. The spore rate was analyzed using a generalized linear model with a binomial distribution. In contrast, we used a linear model for spore dynamics to assess differences in the log-transformed number of spores. In both models, we had regime and day and their interaction as explanatory factors and the replicate as a random factor. Further details can be found in the "statistical analyses" section of

the materials and methods. Table D. Virulence metrics and decomposition. We tested for differences due to parasite regime in virulence, using several host fitness metrics (Model 4a-d). For each of them, we run a linear model with the parasite regime as a fixed factor and replicate it as a random factor. In the presence of a significant effect, we run a posthoc to identify pairwise differences between parasite regimes. After that, we used the maximum hazard from the longevity curves as the main virulence measure and (since the average maximum hazard day was 20) the spore density on days 15 and 18. Spore density was calculated as the individual spore load at days 15 and 18, adjusted by individual wing length. Virulence decomposition analysis was done as in [6]. In brief, exploitation was measured as the natural log-transformed mean spore density from days 15 and 18 for a given regime and replicate, while per parasite pathogenicity was given as the slope of the relationship between natural log-transformed maximum hazard and (i.e., virulence) and exploitation. Lastly, post hoc multiple comparisons were performed with "emmeans", using the default Tukey adjustment across the different models. Infectivity is previously defined as the number of parasites released from the first host to the next stage, which can be the outside environment or a new host, according to Silva *et al.* (2025) [13]. Hence, the chronic parasite load of each individual was considered a proxy of infectiousness as it reflects the amount of parasite that will be transferred to the environment in case of host death (particularly in water). The relationship between infectiousness and virulence was then analyzed using a linear model with infectiousness (or mean chronic load at days 15 and 18 of adulthood) as the dependent variable and the log-transformed maximum hazard of every parasite line as the independent variable. Further details can be found in the "statistical analyses" section of the materials and methods. Fig A. Host developmental traits per replicate. **(a)** Larvae mortality across regimes and replicates. Sample size as follows: Early with 357, 441, 491, 428 and 351: Late with 422, 427, 344, 423 and 395; Stock with 404, 484, 441, 401 and 307 for replicate 1–5 respectively. **(b)** Pupae mortality for each regime and replicate. Missing bars represent replicates with no pupae mortality. Sample size was: Early with 335, 419, 467, 401 and 333: Late with 395, 395, 320, 395 and 368; Stock with 383, 455, 420, 385 and 293 for replicate 1–5 respectively. **(c)** Early pupation, as the proportion of individuals that pupated until day eight post eclosion. Sample size was the following: Early with 330, 416, 465, 398 and 325: Late with 395, 395, 320, 394 and 363; Stock with 381, 452, 419, 385 and 292 for replicate 1–5 respectively. **(d)** Body size at adult emergence using wing length of five-day-old females as a proxy. Each replicate consisted of 10 individuals, illustrated by a different dot shape. *Note*: Every supporting data file is accompanied by a sister *readme file* that describes and explains the variables and information in each column.
(DOCX)

**S1 Data. Host survival data after selection.**
(XLSX)

**S1 File. *Readme* file for S1 Data.**
(TXT)

**S2 Data. Maximum hazard data after selection.**
(XLSX)

**S2 File. R*eadme* file for S2 Data.**
(TXT)

**S3 Data. Host fecundity data after selection.**
(XLSX)

**S3 File. R*eadme* file for S3 Data.**
(TXT)

**S4 Data. Host developmental traits data after selection.**
(XLSX)

**S4 File.** *Readme* file for S4 Data.
(TXT)

**S5 Data.** Host wing size data after selection.
(XLSX)

**S5 File.** *Readme* file for S5 Data.
(TXT)

**S6 Data.** Host spore production rate and infection dynamics data.
(XLSX)

**S6 File.** R*eadme* file for S6 Data.
(TXT)

## Acknowledgments

We thank Tiago G. Zeferino and Gwendoline Acerbi for their advice and technical support.

## Author contributions

**Conceptualization:** Luis M. Silva.

**Data curation:** Luis M. Silva.

**Formal analysis:** Luis M. Silva.

**Funding acquisition:** Jacob C. Koella.

**Investigation:** Luis M. Silva.

**Methodology:** Luis M. Silva, Jacob C. Koella.

**Project administration:** Luis M. Silva, Jacob C. Koella.

**Resources:** Jacob C. Koella.

**Software:** Luis M. Silva.

**Supervision:** Jacob C. Koella.

**Validation:** Luis M. Silva, Jacob C. Koella.

**Visualization:** Luis M. Silva, Jacob C. Koella.

**Writing – original draft:** Luis M. Silva.

**Writing – review & editing:** Jacob C. Koella.

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
