## [Decision Letter · Decision Letter 0]

Complex interactions in the life cycle of a simple parasite shape the evolution of virulence

PLOS Pathogens

Dear Dr. Silva,

Thank you for submitting your manuscript to PLOS Pathogens. After careful consideration, we feel that it has merit but does not fully meet PLOS Pathogens's publication criteria as it currently stands. Therefore, we invite you to submit a revised version of the manuscript that addresses the points raised during the review process.

Please submit your revised manuscript within 60 days Apr 28 2025 11:59PM. If you will need more time than this to complete your revisions, please reply to this message or contact the journal office at plospathogens@plos.org. Please include the following items when submitting your revised manuscript:

We look forward to receiving your revised manuscript.

Kind regards,

Robert L. Unckless, Ph.D.

Academic Editor

PLOS Pathogens

Francis Jiggins

Section Editor

Editor-in-Chief

PLOS Pathogens

Michael Malim

Editor-in-Chief

PLOS Pathogens

orcid.org/0000-0002-7699-2064

**Additional Editor Comments :**

Three reviewers have evaluated your manuscript entitled "Complex interactions in the life cycle of a simple parasite shape the evolution of virulence". All three are generally positive about the results and the contribution to the field. However, each reviewer points out important perceived shortcomings that should be addressed upon resubmission. There was some disagreement among reviewers about the clarity of presentation, but I think this in itself warrants a careful reworking to be sure that results and (importantly) methodology are presented as clearly as possible for a broad audience. Reviewer 1 raises an important point about experimental design and the distinction between sustained host exploitation and increased virulence (Reviewer 3 has a similar comment stated somewhat differently about selection regimes). Reviewers 2 and 3 comment about the results going in different directions at different life stages and both have some suggestions for how to simplify and/or clarify that presentation. Reviewer 2 also has important concerns about controls being presented.

**Journal Requirements:**

1) Please provide an Author Summary. This should appear in your manuscript between the Abstract (if applicable) and the Introduction, and should be 150-200 words long. The aim should be to make your findings accessible to a wide audience that includes both scientists and non-scientists. Sample summaries can be found on our website under Submission Guidelines:

https://journals.plos.org/plospathogens/s/submission-guidelines#loc-parts-of-a-submission

3) Please upload a copy of Figure 4d which you refer to in your text on page 9. Or, if the figure is no longer to be included as part of the submission please remove all reference to it within the text.

**Comments to the Authors:**

**Please note that one of the reviews is uploaded as an attachment.**

**Reviewers' Comments:**

Reviewer's Responses to Questions

**Part I - Summary**

Reviewer #1: The manuscript describes a study of evolution of virulence in mosquito-infecting microsporidian Vavraia culicis in response to short vs. long within-host development time. The authors counducted an experimental evolution trial and show that late-selected parasite lines induce higher host mortality, more rapid development inside the host and higher spore burden in the host; these spore lines are much more efficient in host exploitation.

I think this manuscript is of high quality. The reasoning for the study is well described and addresses important subject in the field of disease ecology and evolution, the size of the experiment is impressive, and the data collected are extensive. The quality of the writing is very high and proves the impressive expertise of the authors. The reasoning in the discussion is fairly complex but described in understandable way. Overall, I value this manuscript very highly. Having that said, I have one major concern, that comes from either a flaw in the experimental design, or imperfect study description. Either way, I think my concerns can be addressed with some changes in the pitch or in study/methods description. The major concern is described below.

Reviewer #2: (No Response)

Reviewer #3: The manuscript presents a study on the evolution of virulence in Vavraia culicis infecting Anopheles gambiae. It provides valuable insights into how transmission timing influences parasite evolution, challenging classical trade-off models. However, clarity in statistical explanations, smoother transitions between sections, and a clearer statement of hypotheses would enhance readability.

Additionally, the logical contradiction to what is expected that hosts dying faster “contain” spores that are less virulent ( and vice-versa for late transmission), may need an experimental approach as it is too central to the manuscript to rely solely on the discussion section.

**Part II – Major Issues: Key Experiments Required for Acceptance**

Reviewer #1: I think this experiment might have one important confounding factor, and I really hope that the authors can show that my worries here are incorrect:

The purpose of the study was to induce selection on virulence through a regulated time of transmission. The authors were perpetually using spores from early-dying mosquitoes or late-dying mosquitoes for 6 generations, and then estimated the virulence related parameters in the parasite and the host. This design does two things: lets the spores develop inside the host for shorter vs longer time, but also it selects spores produced in short-living vs. long-living hosts. The latter outcome of this design is what I’m worried about, because it possibly means that the authors obtained the spores produced in low-quality host (which died soon after infection) vs. high-quality host (that withstood infection for much longer). If that was the case, one could hypothesize that spores produced by low quality host would be of low quality themselves, hence less virulent (as this study shows). The low-quality spores would likely develop inside the host less rapidly (Fig. 3) and induce lower mortality (Fig. 2). Using culling regime on random sub-group of mosquitos instead of letting the mosquitos die of infection would probably allow avoiding the low-quality host bias.

Looking at the design of the experimental evolution here one would naturally expect that parasites killing their host fast would be more virulent and those that let their host live long are less virulent, especially if we consider risk of host mortality as the proxy for virulence. The authors explain that the counterintuitive results stem from the mechanism of prolonged competition of parasite genotypes inside the long-living hosts, resulting in selection for higher virulence. However, I suspect that it might rather be a selection for more sustainable host exploitation than increased virulence (please see Lindsay et all 2023 https://onlinelibrary.wiley.com/doi/10.1111/ele.14218). Selection for more efficient host exploitation would result in much higher spore burden inside the host (Fig. 3b of the manuscript) and much higher host exploitation (Fig. 4a) but not higher per spore virulence (Fig. 4b).

I would greatly appreciate if the authors could address my concerns, both in the response to this review and in the manuscript, because even if these concerns are not due and come from my misunderstanding, same mistakes could be made by other readers.

Reviewer #2: (No Response)

Reviewer #3: General comments:

- is there any prediction of how many parasite replication cycles exist between the early and the late-selected regimes? because according to figure 4b, it seems like the progression of parasite replication within hosts is equivalent for both regimes, which means they would have to differ in initial inocula ...

- results in figure 1 seem to indicate that the parasite basically becomes better adapted to resist the host's immune response - which would somewhat explain the results of fecundity...

- is the parasite dependent on metamorphosis of the host to properly replicate? otherwise, increasing virulence during larval stage would make sense, for a "blindly" more virulent pathogen... this is not what they show, as they did not find any effect on larval mortality.

Saying that the late-selected parasites fasten the host's development seems like a stretch, especially as there is no comparison to the unselected parasite regime...

- in fig 3b, why are loads only detected at day 1 for the late-selected regime, wheres for the early and unselected regimes, loads start at day 2? that could underlie different initial inocula... This could be explained by better adaptation to the host's internal environment, which apparently is what is concluded later: that more spore loads lead to higher virulence...!

ALL IN ALL, I am unsure if it is strictly correct to call the late-selected regime, a selection of delayed transmission, and not merely a better adaptation to the host... according to Michalakis et al (https://pmc.ncbi.nlm.nih.gov/articles/PMC3352405/),

this parasite is not desiccation resistant, meaning that transmission would always be higher in a context where hosts remained close to the (larval) aquatic environment, so assuming that hosts that die from infection "land in water and are transmitted" might be too much of a simplification for the conclusions being made.

specific/methodological issues:

- the 1st timepoint in fig3 (a and b) showing that there are detectable spore counts in the late-regime but not in the other two should be addressed and discussed as it could deeply impact the interpretations and conclusions.

- what is called (when mentioning decomposition of virulence) the "per-parasite pathogenicity" is never directly tested, as only the other arm of virulence, "exploitation", is measured. Additionally, more than once, authors mention that this partition of virulence comprises the action of toxins, but: is it established that this microsporidian produce toxins?

- could it make sense to talk about "host terminal investment" induced by the parasite selected for late-coisas?

- in fig 2c (at least) the comparison with the unselected pathogen should be made

- again, in fig 3a/b, I reiterate that the main conclusions drawn by authors can be due to the confounding effect of the late-selected regime having higher initial inocula... this is probably not a technical problem, as it is standardize for the number of spores exposed to larval stages, but if the parasite has become better adapted at attaching/penetrating host cells, that could lead to that initial difference...

- It was unclear to me if the parasite used has genetic variability or the system is driven solely by mutation. Maybe this could be clarified.

**Part III – Minor Issues: Editorial and Data Presentation Modifications**

Reviewer #1: Additional specific comment, that should be addressed, should the second round of reviews happen:

Lines 387-392: I’m having a hard time figuring out sample sizes and the general flow of the experiment, given that some of the mosquitos are sacrificed for spore counts, some were left to track lifespan, and some were used to estimate reproduction, and the sample sizes are not provided for every parameter measured. I think if this section was reorganized a little bit and sample sizes were added, it would be easier for the reader to keep track of the experimental design.

Reviewer #2: (No Response)

Reviewer #3: line 80: “which IS governed” not “are”

lines 88-90: The sentence “Hence, …” is very obscure. Consider re-writing.

PLOS authors have the option to publish the peer review history of their article (what does this mean? ). If published, this will include your full peer review and any attached files.

**Do you want your identity to be public for this peer review?** For information about this choice, including consent withdrawal, please see our Privacy Policy .

Reviewer #1: No

Reviewer #2: No

Reviewer #3: No

**Figure resubmission:**

**Reproducibility:**



---

## [Decision Letter · Decision Letter 1]

Dear Dr. Silva,

We are pleased to inform you that your manuscript 'Complex interactions in the life cycle of a simple parasite shape the evolution of virulence' has been provisionally accepted for publication in PLOS Pathogens.

Best regards,

Robert L. Unckless, Ph.D.

Academic Editor

PLOS Pathogens

Francis Jiggins

Section Editor

PLOS Pathogens

Sumita Bhaduri-McIntosh

Editor-in-Chief

PLOS Pathogens

orcid.org/0000-0003-2946-9497

Michael Malim

Editor-in-Chief

PLOS Pathogens

orcid.org/0000-0002-7699-2064

Two experts in the field have reviewed the resubmission of "Complex interactions in the life cycle of a simple parasite shape the evolution of virulence". Both were satisfied with the revisions and are enthusiastic about the piece being published.

Reviewer Comments (if any, and for reference):

Reviewer's Responses to Questions

**Part I - Summary**

Reviewer #1: The authors addressed my comments in satisfactory way. I was already impressed with the manuscript before the reviews, therefore I have nothing else to add. Congratulations!

Reviewer #3: The authors have addressed my questions and requests in a acceptable way. It is my opinion that the manuscript deserves publication in its present form, mostly because of the general message it conveys that the data illustrates in a sufficient manner. Specifically, this paper provides an experimental body that highlights the necessity of considering with much more care ecological and life-history traits across the entirety of the life cycle when studying transmission-virulence.

**Part II – Major Issues: Key Experiments Required for Acceptance**

Reviewer #1: (No Response)

Reviewer #3: (No Response)

**Part III – Minor Issues: Editorial and Data Presentation Modifications**

Reviewer #1: (No Response)

Reviewer #3: (No Response)

PLOS authors have the option to publish the peer review history of their article (what does this mean? ). If published, this will include your full peer review and any attached files.

**Do you want your identity to be public for this peer review?** For information about this choice, including consent withdrawal, please see our Privacy Policy .

Reviewer #1: No

Reviewer #3: No

---

## [Editor Report · Acceptance letter]

Dear Dr. Silva,

We are delighted to inform you that your manuscript, "Complex interactions in the life cycle of a simple parasite shape the evolution of virulence," has been formally accepted for publication in PLOS Pathogens.

Best regards,

Sumita Bhaduri-McIntosh

Editor-in-Chief

PLOS Pathogens

orcid.org/0000-0003-2946-9497

Michael Malim

Editor-in-Chief

PLOS Pathogens

orcid.org/0000-0002-7699-2064